# Glaucoma Surgery During Non-Pandemic vs. Pandemic Periods in a Tertiary Center in Romania

**DOI:** 10.3390/medicina61112009

**Published:** 2025-11-10

**Authors:** Nicoleta Anton, Ionuț Iulian Lungu, Francesca Cristiana Dohotariu, Roxana Elena Ciuntu, Ana Maria Picioroagă, Maria Drăgan

**Affiliations:** 1Grigore T. Popa University of Medicine and Pharmacy Iasi, 16 Universității Street, 700115 Iasi, Romania; anton.nicoleta1@umfiasi.ro (N.A.); roxana-elena.ciuntu@umfiasi.ro (R.E.C.); maria.wolszleger@umfiasi.ro (M.D.); 2Ophthalmology Clinic, Sf. Spiridon Emergency Clinical Hospital, 700111 Iasi, Romania; frances7cd@gmail.com (F.C.D.); anapicioroaga@yahoo.com (A.M.P.)

**Keywords:** trabeculectomy, pandemic period, peripheral iridectomy, mitomycin C, neovascular glaucoma, pseudo exfoliative glaucoma, success rate

## Abstract

*Background and Objectives*: This study aims to comparatively evaluate the outcomes of glaucoma surgeries performed by a single surgeon during the non-pandemic period (2019, 2021, and the first quarter of 2022) versus the pandemic year (2020). The analysis focuses on key surgical outcomes, including intraocular pressure (IOP) reduction, intraoperative and postoperative complications, surgical success and failure rates, and their broader clinical implications. *Materials and Methods*: Out of a total of 66 patients admitted between November 2019 and March 2022, 45 met the inclusion and exclusion criteria and were enrolled in the study. All patients underwent glaucoma surgery conducted by the same surgeon employing a standardized technique (trabeculectomy ± iridectomy ± mitomycin C). The evaluated clinical parameters included preoperative and postoperative IOP values (with specific assessment on the first postoperative day), early and late intraoperative and postoperative complications, as well as postoperative success and failure rates. *Results*: The majority of glaucoma cases—particularly those of primary open-angle glaucoma—were recorded in 2021 and 2022, in contrast to 2019 and 2020, when pseudoexfoliative and secondary closed-angle glaucomas predominated. Over the observation period, retrobulbar anesthesia was more frequently utilized in 2019. Statistical analysis indicated that the surgical failure rate was not significantly influenced by the presence of complications, patient age, associated comorbidities, or the specific surgical variant performed. *Conclusions*: The conduct of glaucoma surgery during the pandemic period was marked by substantial operational and clinical constraints when compared to non-pandemic years, primarily as a consequence of decreased patient accessibility and the reprioritization of healthcare resources, despite the acknowledged emergency status of these procedures. Nonetheless, the overall incidence of early intraoperative and postoperative complications remained minimal, with transient intraocular hypotony emerging as the predominant adverse event, observed in ten cases. Across all study cohorts, more than 80% of patients achieved qualified surgical success, while only 18% exhibited surgical failure, underscoring the robustness of standardized operative protocols under variable healthcare conditions. Consistent with the directives of the American Academy of Ophthalmology (AAO) and the European Glaucoma Society (EGS), glaucoma must be regarded as a genuine ophthalmic emergency necessitating prompt surgical intervention when intraocular pressure cannot be adequately managed through pharmacological or laser-based therapies. The current findings reinforce the imperative of timely glaucoma surgery, irrespective of pandemic or non-pandemic circumstances, as a critical measure for averting irreversible optic nerve damage, mitigating functional visual loss, and sustaining long-term ocular integrity.

## 1. Introduction

The COVID-19 pandemic required an unprecedented and rapid adaptation of global healthcare systems in the face of extraordinary adversity. According to data reported by the World Health Organization, as of 20 May 2021, a total of 164,523,894 confirmed cases and 3,412,032 deaths had been recorded worldwide, underscoring the profound magnitude of its impact on humanity. Although the disease primarily manifested as a severe acute respiratory syndrome, SARS-CoV-2 infection also demonstrated multisystemic involvement, generally with favorable outcomes. However, in elderly individuals and in pediatric patients with pre-existing comorbidities, the infection frequently led to severe complications necessitating urgent medical intervention [1,2]. To mitigate the transmission of this highly contagious airborne virus, several preventive strategies proved essential. These included maintaining physical distancing, adherence to mask-wearing protocols, rigorous hand hygiene and disinfection practices, observance of respiratory etiquette, and, most critically, the dissemination of accurate public health information regarding the nature and prevention of the disease [3,4]. During the pandemic, elderly patients with glaucoma encountered significant barriers to care. Visual field constriction and multiple comorbidities hindered their ability to attend local ophthalmologic services and undergo regular IOP monitoring. In an attempt to reduce viral transmission and optimize limited healthcare resources, numerous ophthalmological societies worldwide issued clinical guidelines distinguishing between urgent and non-urgent procedures. Consequently, many elective or non-urgent surgical interventions were deferred [5,6]. Moreover, a substantial number of ophthalmology departments were repurposed into COVID-19 units, admitting exclusively SARS-CoV-2-positive patients and thereby restricting access to ophthalmic services for non-infected individuals [3,4,5,6,7].

Reports from multiple international studies have indicated a marked decline in treatment adherence among patients with Primary Open-Angle Glaucoma (POAG) during the pandemic. Non-adherence was documented in diverse settings, including India [8], Croatia [9], the United States [10], Egypt [11], and Pakistan [12]. The enforcement of strict lockdown measures contributed significantly to reduced adherence rates, attributable to factors such as limited availability and accessibility of medications [8,13,14,15], travel restrictions [8], financial hardship arising from the pandemic’s economic repercussions [8,11], and insufficient awareness or understanding of glaucoma management [8,10,11].

In India, trabeculectomy and bleb-related procedures represented the most frequently performed glaucoma surgeries during the pandemic, with secondary glaucoma accounting for the majority of surgically treated cases (62.5%). In the post-pandemic period, the frequency of minimally invasive glaucoma surgeries (MIGS) increased, while the number of trabeculectomies declined [4,16]. Similarly, in the United Kingdom, although trabeculectomy remained a reference standard, its utilization decreased during the pandemic due to the high frequency of required postoperative assessments and follow-up interventions. Consequently, alternative therapeutic modalities—such as conventional laser therapy, micropulse diode laser, glaucoma drainage devices, deep sclerectomy, and Preserflo microshunt implantation—gained wider adoption [7].

In Romania, a comparable pattern emerged: numerous private ophthalmologic practices and clinics temporarily suspended their operations, while many public hospitals either closed or significantly reduced their surgical activity. This reorganization of healthcare delivery led to a pronounced decline in ophthalmic surgical volume nationwide.

Against this background, the present study aims to evaluate the outcomes of glaucoma surgery performed by a single surgeon between October 2019 and March 2022, comparing the pandemic and non-pandemic periods in a tertiary referral hospital in the Moldova region, the only institution that remained continuously operational throughout the COVID-19 crisis. The study specifically assesses the clinical effects of antiglaucoma surgery (trabeculectomy), with emphasis on intraocular pressure reduction, intra- and postoperative complications, postoperative success and failure rates, and the broader implications associated with surgical failure.

## 2. Materials and Methods

Patient Selection. The present study included patients diagnosed with various forms of glaucoma—inflammatory, neovascular, pseudoexfoliative, and post-traumatic—who were admitted under emergency circumstances and underwent surgical intervention by the same surgeon (Anton Nicoleta) during the non-pandemic period (2019, 2021, and the first quarter of 2022) and the pandemic year (2020). All surgical procedures consisted of trabeculectomy, performed with or without iridectomy (±IP), mitomycin C (±MMC), and tenonectomy, depending on the individual clinical context. Each patient was evaluated both preoperatively and postoperatively according to a standardized institutional protocol.

Inclusion criteria. All emergency-admitted glaucoma cases requiring surgical management were included, provided that trabeculectomy was the procedure of choice.

Exclusion criteria. Patients presenting with phacomorphic glaucoma who required an alternative surgical approach—specifically lens extraction instead of trabeculectomy—were excluded from the study.

Clinical Parameters and Data Collection

The following clinical variables were systematically assessed:

Intraocular pressure (IOP): preoperative, first postoperative day, and one-month follow-up;

Visual acuity (VA): baseline and postoperative values;

Intraoperative and postoperative complications, classified as early or late;

Postoperative success and failure rates according to standard outcome definitions.

Additional data collected included the type of anesthesia administered (general vs. topical or retrobulbar injectable anesthesia), whether surgery was performed on a monocular (single functional) eye, and the number of antiglaucoma medications prescribed preoperatively and postoperatively.

### 2.1. Preoperative Preparation

The choice of anesthesia—local or general—was individualized based on multiple clinical and psychosocial factors, including single-eye status, patient age (younger vs. elderly), systemic comorbidities, and anxiety level. Preoperative evaluation comprised a comprehensive series of laboratory and clinical assessments: complete blood count, liver function tests, urea and creatinine levels, electrolyte panel, and coagulation profile. A cardiology consultation was systematically performed to confirm the patient’s cardiovascular fitness for surgery.

Particular attention was given to anticoagulant therapy, which was discontinued prior to the intervention to minimize intraoperative bleeding risk. In cases scheduled for general anesthesia, chest radiography was additionally required as part of the pre-anesthetic work-up.

General anesthesia was primarily indicated in cases involving a single functional eye, in highly anxious patients, or when there existed a significant risk of intraoperative hemorrhage, to ensure optimal surgical safety and patient comfort.

### 2.2. Surgical Technique

All surgical procedures were conducted according to a standardized trabeculectomy protocol (Table 1).

Localization: The initial trabeculectomy was consistently performed at the 12 o’clock position. Subsequent interventions, where applicable, were localized either superonasally or superotemporally, depending on conjunctival integrity and surgical history.

Use of Mitomycin C (MMC): When adjuvant antimetabolite therapy was indicated, mitomycin C (MMC) was applied at a concentration of 0.2 mg/mL. The solution was administered using a sponge soaked in MMC, placed between the conjunctiva and sclera for an average exposure time of three minutes. Upon removal, the treated area was irrigated thoroughly with balanced salt solution (BSS) to ensure complete elimination of residual MMC and minimize cytotoxic effects.

All surgical interventions were performed by the same experienced ophthalmic surgeon (Anton Nicoleta), following standardized preoperative preparation protocols. Prior to surgery, each patient underwent a comprehensive consultation that included a detailed discussion of the operative technique, potential intraoperative and postoperative complications, postoperative care recommendations, and anticipated visual and functional outcomes. Informed consent was obtained in writing from all participants after ensuring full understanding of the surgical procedure and its potential risks and benefits, in accordance with institutional and ethical standards.

### 2.3. Statistical Analysis

The data were compiled into an SPSS 18.0 database and analyzed using the statistical functions appropriate to each variable, with a 95% significance level. The ana-lytical methods employed included: the ANOVA (analysis of variance) test; Stu-dent’s *t*-test; the χ^2^ test; the Kruskal–Wallis test (a non-parametric test used to compare three or more groups); the Pearson correlation coefficient (r) to assess correlations between variables; the receiver operating characteristic (ROC) curve to evaluate the sensitivity–specificity balance as a prognostic factor; and logistic regression (multivariate analysis).

The ANOVA test was employed to determine descriptive statistical parameters, including minimum, maximum, mean, median, standard deviation, standard error of the mean, and variance. The Kurtosis test (−2 < *p* < 2) was applied to confirm the normal distribution of the dataset and is used when the variable under investigation consists of continuous values.

In determining whether a difference between two means is statistically significant, the t-Student’s test incorporates both the measurement of variability and the weighting of observations, assuming that the series being compared follows a normal distribution.

The Chi-square test and the Likelihood Ratio test are nonparametric procedures used to compare two or more frequency distributions originating from the same population; they are applied when the expected events are mutually exclusive.

The linear regression model is a statistical method that estimates the relationship between a dependent variable and one or more independent variables by fitting a linear equation to the data. It uses this equation to predict outcomes, with the goal of finding the “best fit” line that minimizes the distance between the observed data points and the predicted values.

## 3. Results

Out of a total of 66 patients admitted between November 2019 and March 2022, 45 patients met the established inclusion and exclusion criteria and were consequently enrolled in the study. All participants underwent glaucoma surgery performed by the same surgeon, following an identical operative technique consisting of trabeculectomy with or without iridectomy (±IP) and/or mitomycin C (±MMC), as illustrated in Figure 1.

A total of 21 patients were excluded from the analysis due to a diagnosis of phacomorphic glaucoma, which necessitated an alternative surgical approach—specifically, lens extraction instead of trabeculectomy. In 2019, pseudo exfoliative glaucoma represented the predominant subtype, accounting for 50% of cases. During the pandemic year (2020), neovascular glaucoma became the most frequent diagnosis, comprising 40% of cases. Conversely, in 2021 (45.5%) and 2022 (40%), primary open-angle glaucoma (POAG) was identified as the leading subtype (*p* = 0.067), as illustrated in Figure 2.

Analysis of the distribution by study year revealed that retrobulbar anesthesia (RB) was administered more frequently in 2019 (75%), 2021 (86.4%), and 2022 (60%), with no statistically significant variation observed between years (*p* = 0.323) (Figure 3).

The types of surgical interventions performed are shown in Figure 4. All procedures consisted of trabeculectomy, either as a stand-alone intervention or in combination with peripheral iridectomy (PI), mitomycin C (MMC) application, and/or tenonectomy.

In 2019, 75% of patients, and in 2021 and 2022, 75% and 80%, respectively, had been receiving a combination of three antiglaucoma medications preoperatively. In contrast, 40% of glaucoma cases in 2019 had not received any prior antiglaucoma treatment (*p* = 0.024). Among the four untreated patients, three (75%) were discharged without the need for postoperative antiglaucoma therapy. Similarly, of the 28 patients who had received a triple-drug regimen preoperatively, 21 (75%) were discharged without requiring continued medical treatment (*p* = 0.677) (Figure 5).

Among the associated systemic comorbidities, the most frequently reported were arterial hypertension, diabetes mellitus, and ankylosing spondylitis, the latter being correlated with two cases of inflammatory glaucoma. In a limited subset of cases, 5-fluorouracil (5-FU)—an antifibrotic agent—was administered via subconjunctival injection adjacent to the filtering bleb to reduce postoperative scarring. Ocular massage was prescribed only in exceptional circumstances.

Visual acuity (VA) demonstrated minimal variation in the postoperative period compared to the preoperative baseline values, indicating relative stability of visual function following surgical intervention (Figure 6).

The intraocular pressure (IOP) levels were assessed both preoperatively and postoperatively, revealing a statistically significant postoperative reduction, indicative of effective IOP control and successful aqueous humor outflow restoration following trabeculectomy (Figure 7).

The mean reduction in intraocular pressure (IOP) at discharge demonstrated interannual variation, with the highest mean IOP observed in 2019 (17.0 mmHg) and the lowest in 2022 (12.6 mmHg), reflecting a trend toward improved postoperative pressure control, although the difference did not reach statistical significance (*p* = 0.544) (Figure 8).

Intraoperative and early postoperative complications. Intraoperative complications were documented in three patients in 2020 and two patients in 2021. These included two cases of iridial hemorrhage, hyphema, and secondary vitreous hemorrhage in single functional eyes belonging to patients with concurrent anticoagulant therapy and diabetes mellitus. One additional case involved a suprachoroidal hemorrhage, which necessitated immediate closure of the superficial scleral flap with 9-0 nylon sutures, followed by postoperative cycloplegic therapy, increased oral hydration, and analgesic administration. Early postoperative complications were identified in six patients, three in 2019, two in 2020, and one in 2021—and included hyphema and early failure of the filtering bleb, characterized by bleb flattening and elevated intraocular pressure (IOP). Notably, no complications were reported in 2022, a period during which all surgical interventions were performed under analgosedation or general anesthesia, suggesting enhanced intraoperative stability. Late postoperative complications were recorded in a single case—a 31-year-old female patient from an urban area diagnosed with inflammatory glaucoma—who developed recurrent uveitis associated with moderately elevated IOP.

Reinterventions were required in several instances: two patients with inflammatory glaucoma operated in 2019 underwent subsequent cataract extraction for mature, complicated cataract, which contributed to secondary glaucoma decompensation. Additionally, two patients operated on in 2022 required a repeat trabeculectomy approximately 20 years after their initial procedure (Figure 9).

Success and failure rates. The surgical success rate, defined as maintaining an intraocular pressure (IOP) between 6 and 20 mmHg, was achieved in approximately 85% of cases. Conversely, the failure rate was defined as an IOP below 6 mmHg or exceeding 21 mmHg.

Analysis using a linear regression model demonstrated that the failure rate was not significantly influenced by the presence of complications, sex, age, glaucoma subtype, preoperative IOP, or baseline visual acuity (*p* = 0.719) (Figure 10).

The failure rate was marginally higher among male patients compared to females (60% vs. 47.5%; *p* = 0.597). A notably higher failure rate was observed in the age group over 60 years, reaching 100% (*p* = 0.045). No statistically significant differences were identified with respect to the presence or absence of complications (20% vs. 17.5%; *p* = 0.892) (Figure 11).

The failure rate was notably higher among patients diagnosed with neovascular glaucoma (40%) and pseudoexfoliative glaucoma (40%), although this difference did not reach statistical significance (*p* = 0.169) (Figure 12).

## 4. Discussion

The COVID-19 pandemic posed unprecedented challenges to healthcare systems worldwide, necessitating the implementation of multiple parameters and procedural adjustments to minimize the risk of viral transmission. The evaluation and management of glaucoma patients underwent substantial modifications to ensure the safety of both patients and healthcare professionals. Cross-contamination during ophthalmic examinations was mitigated through the adoption of disposable tonometer tips, thereby reducing potential transmission pathways [17].

Technological adaptations also emerged in the field of visual field assessment, where virtual reality perimetry was introduced as an alternative to conventional automated perimetry. For both glaucoma suspects and confirmed glaucoma patients, imaging modalities—such as optical coherence tomography (OCT) and fundus photography—were increasingly favored over visual field testing, as they offered lower cross-contamination risk, greater ease of disinfection, and faster data acquisition [18]. Given evidence suggesting the presence of SARS-CoV-2 viral RNA in conjunctival secretions, meticulous instrument disinfection remained an essential preventive measure throughout the pandemic [19,20].

Several studies conducted during this period underscored the disruptive impact of the pandemic on the timely evaluation, diagnosis, and treatment of glaucoma, highlighting delayed presentations and progression of disease due to limited access to ophthalmic care.

The present retrospective study analyzed glaucoma surgeries performed at a tertiary referral center in the Moldova region, Romania—the only ophthalmology facility that remained continuously operational during the pandemic, thereby covering an extensive catchment area. According to our findings, the majority of cases treated during the pandemic period were advanced or decompensated glaucomas, reflecting the indirect consequences of systemic healthcare disruptions. This trend was primarily attributable to restricted access to primary care providers, limited availability of reimbursed medications, and interruptions in treatment adherence.

Moreover, the decompensation of systemic comorbidities, such as arterial hypertension and diabetes mellitus, likely contributed to the increased incidence of neovascular glaucoma and proliferative diabetic retinopathy observed during this interval. These findings underscore the complex interplay between systemic disease control, continuity of ophthalmic care, and glaucoma progression in the context of global healthcare crises.

Globally, patterns of healthcare access and surgical practice during the pandemic period differed substantially from those observed in the pre-pandemic years. A large-scale study conducted in India reported that, among 8296 glaucoma surgeries performed during the analyzed interval, secondary glaucoma accounted for the majority of cases requiring surgical intervention (62.5%) during the COVID-19 period. Across all study periods—pre-COVID, COVID, and post-COVID—trabeculectomy and bleb-associated procedures remained the predominant surgical techniques, representing 42.6%, 30%, and 39.8% of all glaucoma operations, respectively. Notably, the frequency of minimally invasive glaucoma surgeries (MIGS) increased in the post-COVID period, while the number of trabeculectomies declined. In parallel, implant-based procedures demonstrated a proportional increase during the pandemic—11.7% pre-COVID, 15% during COVID, and 10.8% post-COVID—a trend correlated with the rising incidence of secondary glaucomas [4]. Another Indian study comparing surgical profiles across consecutive years identified a proportional decrease in incisional glaucoma surgeries (from 60.86% to lower levels) alongside an increase in emergency cataract surgeries (27.15%) and transscleral cyclophotocoagulation (TSCP) procedures, which rose from 8.16% in 2019 to higher levels during 2020. These findings reflect a strategic shift toward shorter, less invasive interventions in response to pandemic-related restrictions and resource reallocation.

In Poland, a retrospective study demonstrated a 50% reduction in glaucoma surgeries during the pandemic compared with an equivalent pre-pandemic interval, accompanied by significant changes in the distribution of surgical techniques. In the pre-pandemic group, the most frequently performed procedures were Ex Press shunt implantation (33.7%) and trabeculectomy (31.5%). In contrast, during the pandemic, trabeculectomy accounted for 50.0% of procedures, followed by Preserflo microshunt implantation (11.6%), iStent placement (8.7%), and transscleral cyclophotocoagulation (TSCP) (8.7%). Collectively, these data suggest that the COVID-19 pandemic was associated with a decline in extensive antiglaucoma surgeries and a corresponding increase in shorter, less complex interventions, reflecting both logistical constraints and a focus on minimizing patient contact time [3].

At a tertiary referral center in Verona, Italy, analysis of surgical volume and procedural distribution during the pandemic year revealed a 30% reduction in glaucoma surgeries compared with the pre-pandemic period. Only 24.3% of interventions were performed under general anesthesia, in contrast to 41.5% prior to the pandemic, reflecting an effort to minimize aerosol-generating procedures and reduce perioperative risks. Notably, the number of surgeries performed on eyes with advanced or end-stage glaucoma nearly doubled during this period. Consistent with the findings of the present study, the majority of patients presented at more advanced stages of disease, a consequence primarily attributed to delayed referrals and restricted access to routine ophthalmologic services. The temporary closure of the operating theater during the first quarter of the pandemic further contributed to the accumulation of advanced and decompensated cases that subsequently required urgent surgical intervention [21].

In Germany, an analysis based on data from the Federal Joint Committee’s (G-BA) national quality reports, encompassing glaucoma surgeries performed in 296 hospitals between 2019 and 2022, demonstrated similar trends. The total number of glaucoma surgeries declined by 8.5% in 2020 compared to 2019 but returned to pre-pandemic levels by 2021. Over the entire four-year observation period, the frequency of traditional trabeculectomies exhibited a steady decline, whereas bleb-forming filtering devices were increasingly utilized. Across all study years, cyclodestructive procedures—particularly transscleral cyclophotocoagulation—remained the most frequently performed interventions, underscoring a global shift toward less invasive, safer, and time-efficient surgical techniques during the pandemic [22].

In the present study, all surgical interventions consisted exclusively of trabeculectomy, performed using a standardized technique. A marked increase in advanced angle-closure glaucoma cases was observed in 2021, accompanied by a higher number of surgeries performed on single functional eyes and cases of lens-induced glaucoma. This trend contrasted with 2020, when neovascular glaucoma represented the predominant subtype. Across all study years, the reduction in IOP following surgery was statistically significant and comparable between groups, with the greatest mean reduction recorded in 2021. The failure rate was significantly higher among patients with neovascular and pseudoexfoliative glaucoma, whereas the overall qualified success rate exceeded 80% in all groups, and only 18% of cases were categorized as surgical failures. The incidence of intraoperative and early postoperative complications was notably low, with early ocular hypotony identified in 10 cases, representing the most frequent complication. These findings align with reports in the literature from the same period, which consistently describe a predominance of secondary, neovascular, and acute glaucomas presenting at advanced stages, frequently necessitating emergency surgical intervention. Before the onset of the COVID-19 pandemic, trabeculectomy was regarded as the standard surgical approach for glaucoma management [16,23,24], with its adoption varying among glaucoma specialists—87% in the United Kingdom [7] and 62.8% in Italy [23].


**Limitations and Perspectives.**


A major limitation of this study is the relatively small sample size, which may constrain the generalizability of the findings. However, perhaps the most significant limitation lies in the fact that all surgical procedures were performed by a single glaucoma specialist. Consequently, the surgical outcomes—including complication rates, surgical success, and failure rates—may have been influenced by the surgeon’s individual technical expertise and operative judgment, introducing a potential source of procedural bias.

Another persistent challenge involves ensuring patient adherence to postoperative follow-up, which is essential for monitoring disease progression and optimizing long-term outcomes. While telemedicine and teleconsultation demonstrated considerable potential in the pre- and post-pandemic periods, their efficacy during the pandemic was constrained by reduced accessibility and infrastructure limitations. During the height of the COVID-19 outbreak, routine outpatient services and elective procedures were largely suspended, and hospital-based ophthalmic care was restricted to emergency cases.

In this context, teleophthalmology—a field that had been expanding globally for over a decade—emerged as an indispensable tool for maintaining continuity of care. Tele glaucoma platforms, in particular, have shown promise for screening [25], diagnostic consultation [26,27], and long-term therapeutic monitoring [27,28,29,30]. Such approaches not only ensure sustained patient surveillance but also protect both patients and clinicians by minimizing direct exposure and reducing hospital visits to only those necessitated by urgent clinical conditions.

## 5. Conclusions

Antiglaucoma surgery encountered a series of substantial challenges during the pandemic period when compared to the non-pandemic years, despite its classification as an emergency procedure. These difficulties primarily arose from restricted access to medical services and a significant decline in healthcare utilization among glaucoma patients requiring surgical intervention.

A notable surge in advanced angle-closure glaucoma cases was documented in 2021, accompanied by an increase in surgeries performed on single functional eyes and in lens-induced glaucomas, contrasting with 2020, when secondary open-angle and primary open-angle glaucomas predominated. The incidence of intraoperative and early postoperative complications remained low, with early ocular hypotony being the most frequent event, recorded in ten cases. Across all study groups, more than 80% of patients achieved qualified surgical success, whereas only 18% were classified as failures.

Successful glaucoma surgery demands a comprehensive understanding of surgical techniques, awareness of potential intraoperative complications, and meticulous preoperative preparation. The selection of the appropriate anesthetic approach—either local or general—should be individualized based on patient-specific factors such as single-eye status, younger age, male sex, and associated systemic comorbidities, as this choice can facilitate surgical precision while minimizing cardiovascular and anesthetic risks.

Although the present study involved a limited number of patients, its findings underscore the marked reduction in healthcare accessibility for glaucoma patients during the pandemic period. This limitation was largely attributable to the scarcity of operational medical facilities and to patients’ apprehension regarding potential viral exposure. Consequently, a pronounced discrepancy emerged in the number of cases presented for diagnosis and surgical treatment within the Moldova region.

Furthermore, the absence of advanced home-based technologies for monitoring IOP and visual field changes—such as home tonometry and home perimetry—coupled with the limited implementation of telemedicine and teleophthalmology services, further hindered the evaluation and follow-up of a broader glaucoma population during this period.

According to the guidelines of the American Academy of Ophthalmology (AAO) and the European Glaucoma Society (EGS), glaucoma constitutes a true ophthalmologic emergency. Surgical intervention is therefore imperative when intraocular pressure remains uncontrolled, irrespective of whether the clinical context is pandemic or non-pandemic, to prevent irreversible functional loss and preserve long-term visual function.

## Figures and Tables

**Figure 1 medicina-61-02009-f001:**
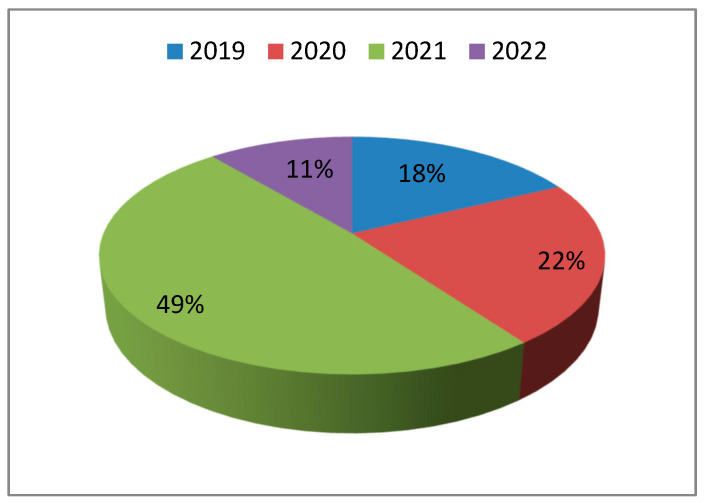
Distribution of the study population by year of surgery. The proportion of patients undergoing glaucoma surgery was 18% in 2019, 22% in 2020, 49% in 2021, and 11% in 2022.

**Figure 2 medicina-61-02009-f002:**
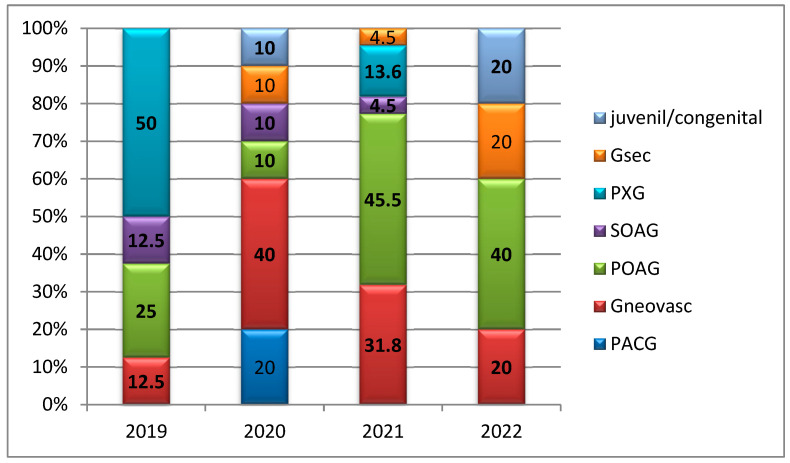
Distribution of glaucoma cases by year of study according to glaucoma subtype. Abbreviations: Gsec—secondary glaucoma; PXG—pseudo exfoliative glaucoma; SOAG—secondary open-angle glaucoma; POAG—primary open-angle glaucoma; PACG—primary angle-closure glaucoma.

**Figure 3 medicina-61-02009-f003:**
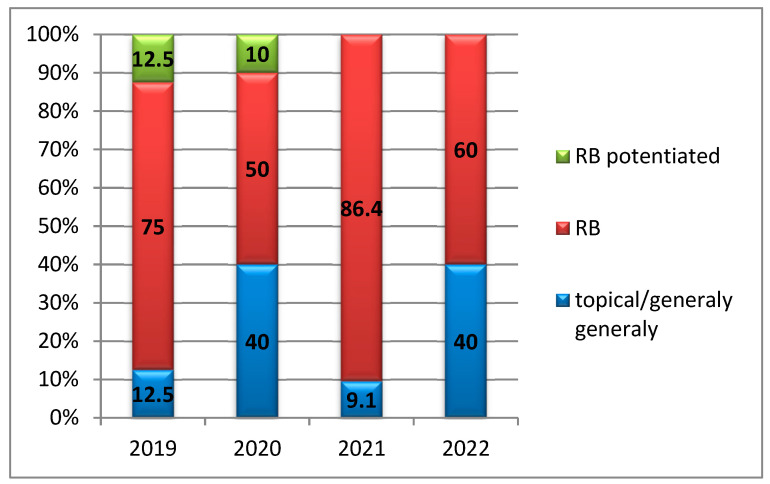
Distribution of anesthesia types by year of study. The anesthesia modalities utilized included potentiated retrobulbar anesthesia, retrobulbar anesthesia alone, topical anesthesia, and general anesthesia, with year-to-year variations reflecting individual patient characteristics and clinical indications.

**Figure 4 medicina-61-02009-f004:**
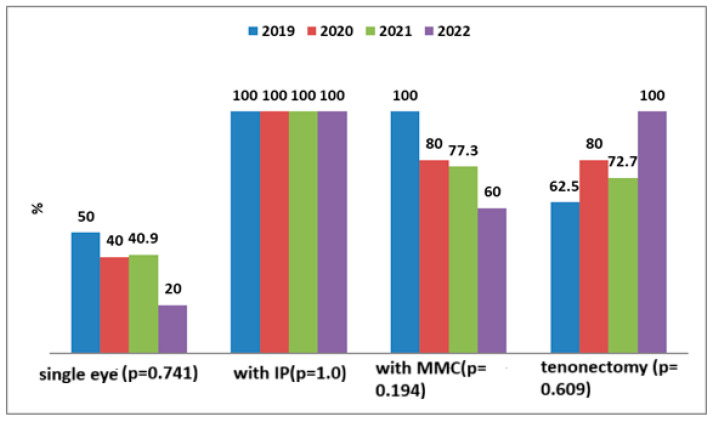
Types of surgical interventions performed. All procedures consisted of trabeculectomy, either as a standalone intervention or in combination with peripheral iridectomy (PI), mitomycin C (MMC) application, and/or tenonectomy. The specific surgical configuration was determined according to the type of glaucoma, the conjunctival status, and intraoperative anatomical considerations.

**Figure 5 medicina-61-02009-f005:**
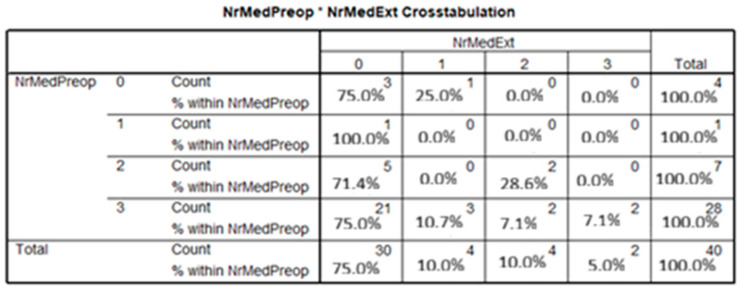
Number of antiglaucoma medications administered across all study years (2019–2022). The distribution of preoperative and postoperative antiglaucoma therapy was recorded and analyzed for all enrolled patients.

**Figure 6 medicina-61-02009-f006:**
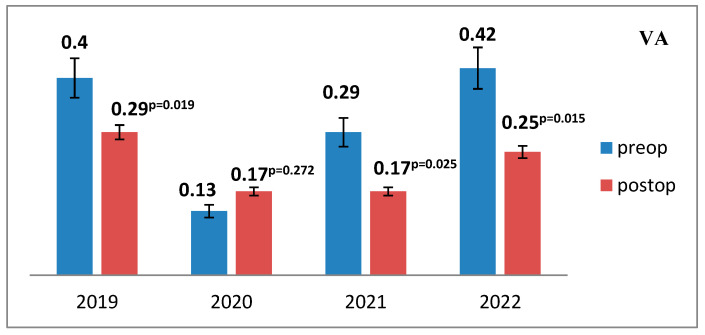
Mean preoperative and postoperative visual acuity values across the study period (2019–2022). The mean visual acuity (VA) values were recorded and compared for each study year, demonstrating overall postoperative stability relative to baseline measures.

**Figure 7 medicina-61-02009-f007:**
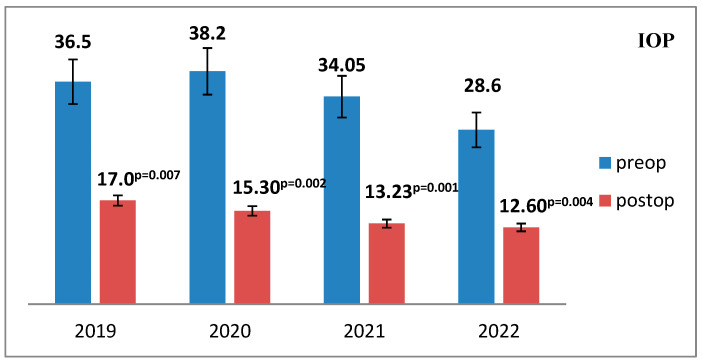
Mean intraocular pressure (IOP) values before and after surgery across the study period (2019–2022). The mean preoperative and postoperative IOP values were recorded and analyzed for each study year to evaluate surgical efficacy in pressure reduction.

**Figure 8 medicina-61-02009-f008:**
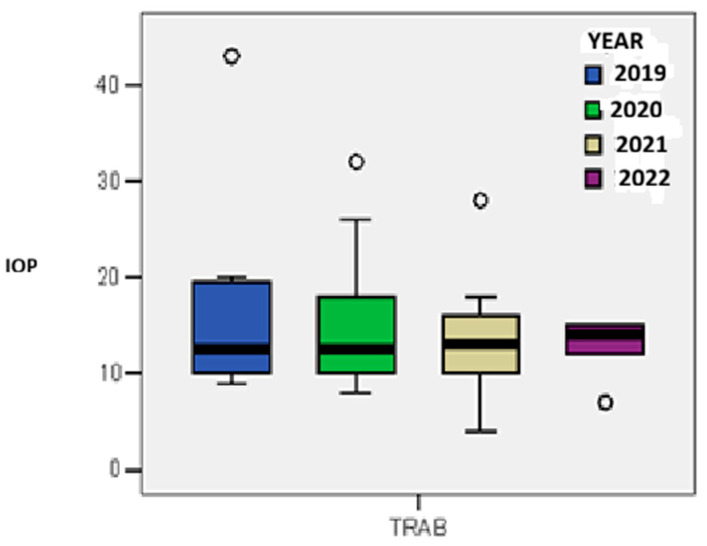
The mean level of pressure reduction at discharge by year of study.

**Figure 9 medicina-61-02009-f009:**
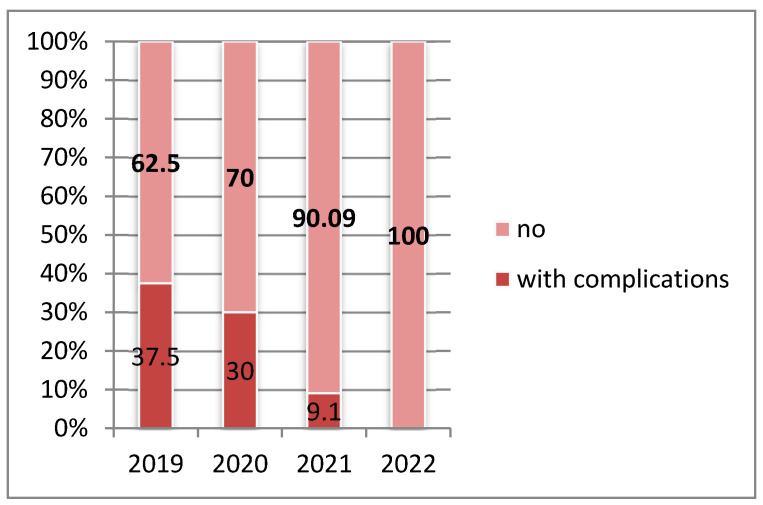
Distribution of glaucoma surgery cases by year of study according to the presence of intraoperative and postoperative complications (2019–2022). A statistically significant variation in complication frequency was observed across the study period (*p* = 0.026).

**Figure 10 medicina-61-02009-f010:**
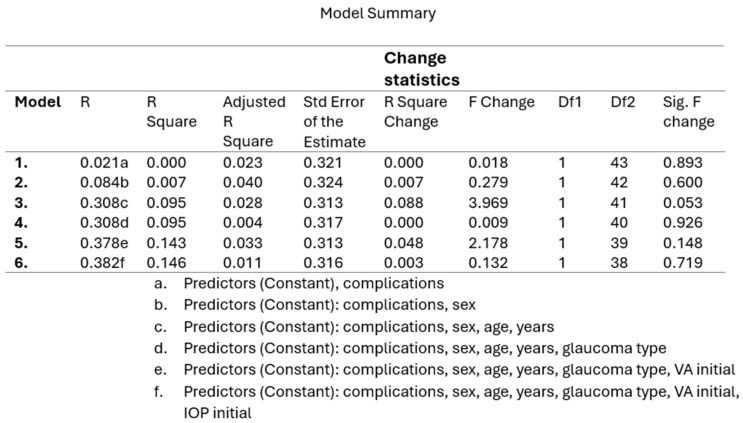
Linear regression model illustrating the predictors of surgical failure. The model evaluates the influence of key variables—presence of complications, sex, age, type of glaucoma, baseline visual acuity, and preoperative intraocular pressure (IOP)—on the overall failure rate following glaucoma surgery.

**Figure 11 medicina-61-02009-f011:**
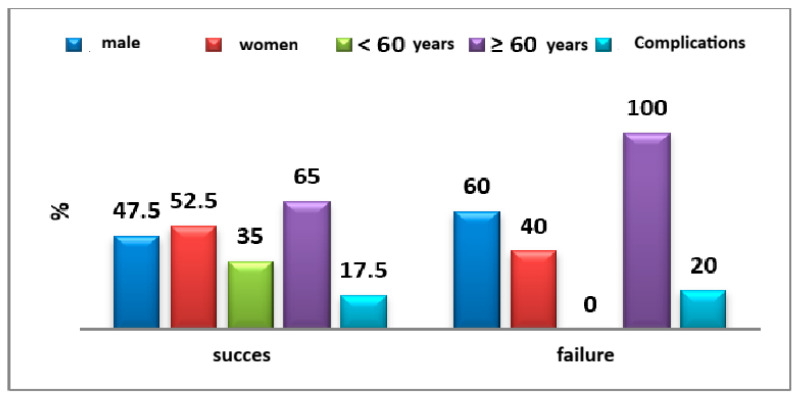
Distribution of success/failure cases according to epidemiological characteristics.

**Figure 12 medicina-61-02009-f012:**
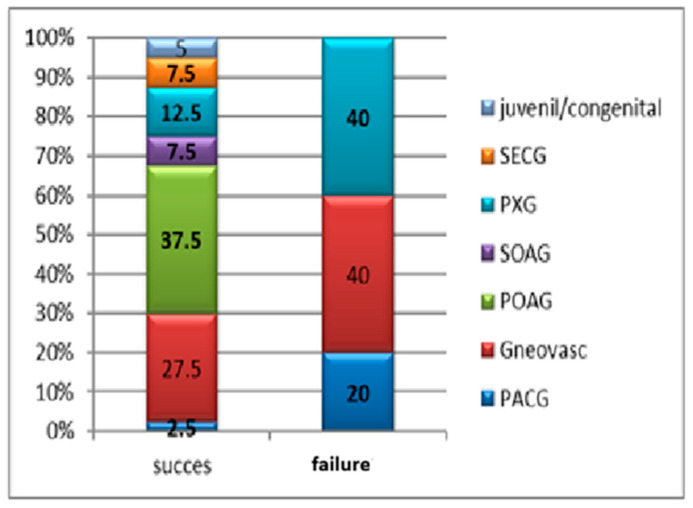
Distribution of surgical success and failure cases according to glaucoma subtype. A higher failure rate was observed among patients with neovascular glaucoma and pseudoexfoliative glaucoma, although the difference was not statistically significant (*p* = 0.169). Abbreviations: Gsec—secondary glaucoma; PXG pseudoexfoliative glaucoma; SOAG—secondary open-angle glaucoma; POAG—primary open-angle glaucoma; PACG—primary angle-closure glaucoma.

**Table 1 medicina-61-02009-t001:** Summary of the standardized trabeculectomy protocol and intraoperative parameters.

Surgical Protocol	Type of Anesthesia Used
Wide fornix-based conjunctival incision, 8 mm in length	General anesthesia (depending on factors such as single-eye status, patient age—young or elderly—associated comorbidities, or anxiety)
Delineation of a superficial scleral flap (50% thickness), rectangular, 4 × 4 mm	Topical/retrobulbar injectable anesthesia
Application of MMC (Mitomycin C—antifibrotic agent)	
Delineation of a deep scleral flap, 3 × 2 mm	
Paracentesis of the anterior chamber (AC), intraocular miotic administration	
Excision of the deep scleral block (3 × 2 mm), peripheral iridectomy	
Suturing of the superficial flap with 5/3 sutures, placement of a releasable suture in selected cases	
Restoration of ocular tone with BSS, assessment of filtration	
Continuous conjunctival suture at the peripheral cornea	

## Data Availability

The datasets used and analyzed in this study are available from the corresponding author on reasonable request.

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
