# Peer review of "Glaucoma Surgery During Non-Pandemic vs. Pandemic Periods in a Tertiary Center in Romania"

_medicina, 2025, doi:10.3390/medicina61112009_

Round 1
Reviewer 1 Report
Comments and Suggestions for Authors
The manuscript provides a comparative analysis of glaucoma surgeries performed between 2019 and 2022, highlighting limitations in timely access to treatment and surgical intervention, primarily due to the impact of the COVID-19 pandemic. The authors present comprehensive data, including the number of glaucoma patients, types of surgeries performed, types of anesthesia used, records of administered antiglaucoma treatments, visual acuity and intraocular pressure measurements, and the success rates of glaucoma operations during the study period. These findings collectively demonstrate reduced healthcare accessibility and increased disease severity due to delayed treatment during the pandemic. Overall, the authors have effectively utilized a wide range of medical records to support their conclusions. However, some portions of the text and several figures were difficult to interpret. I have outlined both major and minor concerns below.
Some concerns and suggestions:
- Lines 25–32: I recommend that the authors summarize the results in a more concise and clarified form. Presenting raw data in this section may overwhelm or confuse readers. It would be clearer and more effective to replace the detailed numerical data with brief interpretations or representative summaries that convey the key findings and their implications.
- Line 91: In the Materials and Methods section, please specify the statistical analysis tools used in this study.
- Line 138: In the Results section, please specify the statistical tests used in the figure legend, if applicable.
- Figures 2, 3, 7, and 8: Please add titles or labels to the y-axes in Figures 2, 3, 7, and 8 to enhance clarity. Additionally, for Figures 7 and 8, the numbers above the red bars appear misaligned. Please also ensure consistent number formatting—for example, replace “17,0” with “17.0”.
- Figure 12: “<60 ani” should be “<60 years old”
- Figure 13: “eÈ™ec” should be “failure”
- Figures 12 and 13: Please add more interpretations in Figures 12 and 13. If these figures represent summary data from 2019 to 2022, the authors should explain how they support the central hypothesis of the manuscript.
- The concluding section contains content that is highly repetitive of the abstract. I recommend summarizing and revising the abstract to emphasize the main findings more concisely and clearly, avoiding redundancy with the conclusion.
Minor suggestions:
- Line 27: The term “closed uncle secondary glaucoma” should be corrected to “closed-angle secondary glaucoma.”
- Lines 37–38: Please clarify the abbreviations “AAO” and “EGS” at their first mention in the manuscript. Although they are listed in abbreviations.
- Lines 51–53: Several words in these lines appear with incorrect line breaks, such as “compli-cations,” “dis-tancing,” and “compli-ance.” Please review the manuscript for consistent formatting and correct any improper word breaks.
- Line 63: (3-6) should be [3-6]
- Line 150: “(lens extraction.” should be “(lens extraction).”
Author Response
Thank you very much for such a thorough evaluation of the article.
We made changes . I added the information in the text marked in yellow.
Thank you very much for your appreciation and we are at your disposal for any other changes.
Kind regards
Nicoleta Anton

Reviewer 2 Report
Comments and Suggestions for Authors
You have presented observational data. The figures show dramatic diferences year to year. ouneed to do a bit more to exlain those differences, making itg clear that you are comment on poosibler explanagtions, NOT proven ones.
Comments on the Quality of English Languagethere are some awkward wordiigs.
Author Response

(The authors gave the same response as above.)

Reviewer 3 Report
Comments and Suggestions for Authors
(A) Provide an overview/summary of the manuscript
The study compared glaucoma surgery outcomes performed by the same surgeon during non-pandemic (2019, 2021, and early 2022) and pandemic (2020) periods. Out of 66 patients, 45 met the inclusion criteria. All underwent surgery with consistent techniques, primarily trabeculectomy with or without adjuncts. Clinical parameters assessed included pre- and postoperative IOP, complications, success, and failure rates. Most cases were from 2021 (38%). Pseudoexfoliative glaucoma was predominant in 2019 (41.7%), while closed-angle secondary glaucoma led in 2020 (61.1%) and neovascular glaucoma in 2020 (40%). In 2021 and 2022, primary open-angle glaucoma was most common. Retrobulbar anesthesia was more frequent in 2019. Failure rates were unaffected by complications, age, comorbidities, or procedure type. Despite pandemic-related challenges, the overall success was over 80%, with 18% failures. The authors emphasized that glaucoma surgery remains urgent regardless of pandemic status, aligning with AAO and EGS guidelines to prevent vision loss.
(B) Introduction and discussion
The authors clearly stated their study objectives, and the data presented support the conclusions. However, the concerns raised in the reviewer's comment (E) should be addressed.
(C) Materials and methods
The methods and statistical analyses employed in this study are generally appropriate; however, the concerns raised in Reviewer's comment (E) should be addressed to ensure clarity and rigor.
(D) Results
The results are generally reliable and valid; however, the concerns raised in Reviewer's comment (E) should be addressed.
(E) Reviewer's comment
The study was confined to evaluating the surgical performance of a single glaucoma surgeon on a limited patient cohort, with patient characteristics potentially influenced by the COVID-19 pandemic.
The following points highlight the primary concerns:
#1. Please explain the methodology used by the authors to determine the appropriate sample size, ensuring the validity of the statistical analysis.
#2. In lines 335-338, the authors state that "A notable increase in advanced angle-closure glaucoma cases was observed in 2021, along with a rise in surgeries on single functional eyes and lens-induced glaucomas, compared to 2020, when secondary open-angle and primary open-angle glaucomas were predominant." This statement appears inconsistent with the earlier statements in lines 26-30, which note, "In 2019, cases of pseudoexfoliative glaucoma predominated (41.7%), while in 2020, cases of closed-angle secondary glaucoma predominated (61.1%). In 2019, pseudoexfoliative glaucoma was the most prevalent subtype (50%), whereas in 2020, neovascular glaucoma predominated (40%). In contrast, in 2021 (45.5%) and 2022 (40%), the most frequent diagnosis was primary open-angle glaucoma." or lines 150-154 state, "In 2019, pseudoexfoliative glaucoma was the most prevalent subtype (50%), whereas in 2020, neovascular glaucoma predominated (40%). In contrast, in 2021 (45.5%) and 2022 (40%) was primary open-angle glaucoma (p = 0.067) (Figure 2)." These conflicting statements should be carefully revised for clarity and accuracy.
#3. The authors should comprehensively acknowledge and address potential biases related to the surgeon's influence, as these factors can substantially affect the validity of the study's findings. Special consideration should be given to the impact of outcomes from a single surgeon within the study. Such biases may arise from various elements, including the types of procedures performed, the surgeon’s level of experience, decision-making processes, and possible conflicts of interest. Furthermore, it is crucial to assess how these biases might influence evaluations of procedure success rates, complication rates, and overall patient outcomes.
#4. The manuscript contains multiple typographical errors and would greatly benefit from a thorough proofreading and professional editing service to enhance its linguistic clarity and overall quality.
Comments on the Quality of English LanguageThe manuscript has numerous typographical errors in English. It is strongly advisable for the authors to have it thoroughly proofread and professionally edited by a language editing service.
Author Response

(The authors gave the same response as above.)

Round 2
Reviewer 3 Report
Comments and Suggestions for Authors
(F) Reviewer's comment on the revised manuscript
The manuscript has been revised in accordance with my recommendations to improve its overall quality and consistency. Nonetheless, the authors are advised to address the following issues.
#1. The manuscript’s Statistical Analysis section is notably deficient and does not uphold rigorous scientific standards. The authors have not provided details regarding the methodology employed for sample size determination, which critically compromises the credibility of the statistical findings. Furthermore, the description offered is merely a broad overview of each analytical technique and omits vital specifics about the particular objectives of each analysis, representing a significant lapse in methodological transparency and scientific rigor.
#2. The manuscript consistently demonstrates numerous typographical errors and necessitates comprehensive proofreading and professional editing to markedly improve its linguistic clarity and overall scholarly quality. The authors are advised to systematically revise all sections of the manuscript.
Comments on the Quality of English LanguageThe manuscript exhibits a substantial number of typographical errors and would benefit significantly from thorough proofreading and professional editing to enhance its linguistic precision and scholarly rigor. Authors are encouraged to undertake a systematic revision of all sections. The subsequent examples serve to illustrate these recurrent issues.
#1. The authors failed to consistently use abbreviations after their initial first appearance and repeatedly abbreviated terms unnecessarily.
#2. There are English typographical errors, such as the misplaced character "._' in line 33.
Round 2
The manuscript has been revised in accordance with my recommendations to improve its overall quality and consistency. Nonetheless, the authors are advised to address the following issues.
#1. The manuscript’s Statistical Analysis section is notably deficient and does not uphold rigorous scientific standards. The authors have not provided details regarding the methodology employed for sample size determination, which critically compromises the credibility of the statistical findings. Furthermore, the description offered is merely a broad overview of each analytical technique and omits vital specifics about the particular objectives of each analysis, representing a significant lapse in methodological transparency and scientific rigor.
#2. The manuscript consistently demonstrates numerous typographical errors and necessitates comprehensive proofreading and professional editing to markedly improve its linguistic clarity and overall scholarly quality. The authors are advised to systematically revise all sections of the manuscript.
Comments on the Quality of English Language
The manuscript exhibits a substantial number of typographical errors and would benefit significantly from thorough proofreading and professional editing to enhance its linguistic precision and scholarly rigor. Authors are encouraged to undertake a systematic revision of all sections. The subsequent examples serve to illustrate these recurrent issues.
#1. The authors failed to consistently use abbreviations after their initial first appearance and repeatedly abbreviated terms unnecessarily.
#2. There are English typographical errors, such as the misplaced character "._' in line 33.
Thank you for your suggestions. We will call for a serious evaluation by specialists of the linguistic quality of the material.
Thank you very much for such a thorough evaluation of the article.
We made changes . I added the information in the text marked in yellow.
Thank you very much for your appreciation and we are at your disposal for any other changes.
Kind regards
Nicoleta Anton